



# Plasma density gradients at the edge of polar ionospheric holes: the presence and absence of phase scintillation

Luke A. Jenner, [1], Alan G. Wood [1], Gareth D. Dorrian [1], Kjellmar Oksavik [2,3], Timothy K. Yeoman [4], Alexandra R. Fogg [4], and Anthea J. Coster [5]

[1] School of Science & Technology, Nottingham Trent University, Nottingham, UK.

[2] Birkeland Centre for Space Science, Department of Physics and Technology, University of Bergen, Bergen, Norway

[3] Arctic Geophysics, University Centre in Svalbard, Longyearbyen, Norway

[4] Department of Physics and Astronomy, University of Leicester, Leicester, UK.

[5] MIT Haystack Observatory, Massachusetts, USA.

*Correspondence to*: Alan G. Wood (alan.wood@ntu.ac.uk)

**Abstract.** Polar holes were observed in the high-latitude ionosphere during a series of multi-instrument case studies close to the northern hemisphere winter solstice in 2014 and 2015. These holes were observed during geomagnetically quiet conditions and under a range of solar activities using the European Incoherent Scatter Scientific Association (EISCAT) Svalbard Radar (ESR) and measurements from Global Navigational Satellite System (GNSS) satellites. Steep electron density gradients have been associated with phase scintillation in previous studies, however, no enhanced scintillation was detected within the electron density gradients at these boundaries. It is suggested that the lack of phase scintillation may be due to low plasma density levels and a lack of intense particle precipitation. It may be that both significant electron density gradients and that plasma density levels above a certain threshold are required for scintillation to occur.

## 1   Introduction

The F-region ionosphere is a weakly ionised plasma in the Earth's atmosphere extending from an altitude of ~150 to ~500 km, above which it merges with Earth's plasmasphere. Large-scale plasma structures with a horizontal extent of tens to hundreds of km are routinely observed in the F-region high-latitude ionosphere (Tsunoda, 1988). One type of structure commonly observed are polar cap patches, also referred to as patches, which are enhancements of plasma density with at least twice the background value and have a horizontal spatial extent of 100 km or greater (Crowley, 1996). Buchau et al. (1983) observed such patches of enhanced ionisation drifting antisunward with the background plasma flow in the central region of the polar cap at Thule, Greenland (77.5° N, 69.2° W; 85.4° MLAT, 32.4° MLON). The patch densities were larger than could be produced due to the observed flux of precipitating particles, and it was concluded that the patches were not produced locally by precipitation. Weber et al. (1984) suggested that the patches



were produced on the dayside at auroral or subauroral latitudes and then convected antisunward to higher, polar

latitudes. A comparison of average maps of the electron density and high-latitude convection pattern suggested that

solar-produced plasma was drawn into the polar cap as a continuous density enhancement known as the Tongue-of-

Ionisation (TOI) (Foster et al., 1984). Several mechanisms have been proposed to break a TOI into a series of patches,

including variations in the high-latitude convection pattern moving flux tubes in and out of sunlight (Anderson et al.,

1988), expansion and contraction of the high-latitude convection pattern in response to transient bursts of reconnection

drawing in plasma from different latitudes (Cowley and Lockwood, 1992; Lockwood and Carlson, 1992; Carlson et al.,

2002, 2004, 2006), variations in the y-component of the Interplanetary Magnetic Field (IMF) drawing in plasma from

different magnetic local times (MLT) (Sojka et al., 1993), variation of the z-component of the IMF altering whether

plasma could be drawn in to the polar cap (Valladares et al., 1998), erosion of plasma densities due to enhanced

recombination during a flow channel event (Rodger et al., 1994; Valladares et al., 1994), and modification of the density

of the photoionised plasma transported into the polar cap by particle precipitation (Walker et al., 1999; Millward et al.,

1999). Patches have been observed travelling thousands of kilometres across the polar regions (Weber, 1986; Oksavik

et al., 2010; Nishimura et al., 2014), and are primarily associated with times when the z-component of the IMF is

negative (Buchau and Reinisch, 1991).

Blobs are also plasma density enhancements, however, unlike patches, they occur outside the polar cap. They are

further categorised into boundary blobs, subauroral blobs, and auroral blobs (Rino, 1983; Jin et al., 2016). Boundary

blobs are found near the equatorward auroral boundary, neighbouring the ionospheric trough's poleward wall.

Parkinson et al. (2002) observed patches leaving the polar cap, slowing in the antisunward direction and then beginning

to move zonally. It was suggested that these patches would form boundary blobs, and this was later confirmed by Pryse

et al. (2006) who compared the plasma density in a polar cap patch to that within a boundary blob which the patch

subsequently formed. Subauroral blobs have a similar appearance to boundary blobs, however, they are found in the

ionospheric trough. Auroral blobs are found within the auroral oval and seem to be longitudinally restricted. The most

likely mechanism for their creation is particle precipitation (Jones et al., 1997).

Not all ionospheric structures are enhancements of the background plasma; polar ionospheric holes are regions of

low plasma density. Brinton et al. (1978) observed a depletion of this kind under conditions of low solar activity

(F10.7=71 sfu) and low magnetic activity ($K_p$ = 2). This depletion was also associated with a minimum of electron

temperatures, indicating the absence of local particle precipitation. Polar holes are generally located between 21 and 06

MLT and 70°-80° magnetic latitude and typically have steep plasma density gradients at their boundaries. They are

believed to be produced when plasma in the high-latitude convection pattern circulates in perpetual darkness. Plasma

loss by recombination in the absence of a plasma source causes density levels to drop. This idea is supported by the

conditions under which polar holes have generally been observed, namely quiet geomagnetic activity ($K_p$ 2 or less)

when the contribution to the plasma densities from particle precipitation will be low (Brinton et al., 1978). The electron

densities inside of the polar holes are seen to reach a minimum in the range of $10^8$-$10^{11}$ electrons·m$^{-3}$ (Obara and Oya,



1989, Benson and Grebowsky, 2001) and, while there is variation between holes, inside of a singular polar hole the density level is very consistent.

Smaller scale structures can arise at steep plasma density gradients due to instability processes such as the gradient-drift instability (GDI) (Keskinen and Ossakow, 1983) and the velocity shear driven instability (Kelvin-Helmholtz instability, KHI). Carlson et al. (2008) proposed and that the real process involves both mechanisms acting on different time scales. The smaller scale (tens of meters to tens of kilometers) structures that arise can cause rapid phase and amplitude fluctuations in radio signals, in a phenomenon known as ionospheric scintillation. Since the second world war, large numbers of studies have shown the effect of ionospheric irregularities on radio signals, as reviewed by Aarons (1982). The morphology of these irregularities has been extensively studied at high-latitudes (e.g. Kersley, 1972), together with the effects upon the propagation of radio signals in this region (e.g. Kersley et al., 1995).

More recently studies have focussed on Global Navigation Satellite System (GNSS) frequencies, where scintillation poses a substantial threat to the integrity, availability and accuracy of GNSS positioning, leading to positioning errors and service outages due to signal tracking problems at the GNSS receiver. A direct connection between gradients in the Total Electron Content (TEC) at the edge of a plasma stream with both phase and amplitude scintillation has been observed (Mitchell et al., 2005) and plasma structuring caused by auroral precipitation has been linked to the loss of signal lock by a GNSS receiver (Elmas et al., 2011; Smith et al., 2008; Oksavik et al., 2015). A statistical study has shown an agreement between both phase and amplitude scintillation with the asymmetric distribution of polar cap patches around magnetic midnight (Spogli et al., 2009) and that auroral emissions correlate with GNSS signal phase scintillation (Kinrade et al., 2013; van der Meeren et al., 2015). Phase and amplitude scintillation can be associated with the larger spatial structures associated with polar cap patches (Alfonsi et al., 2011). Phase scintillation is usually the dominant process at high latitudes (Spogli et al., 2009; Prikryl et al., 2015) and this is the focus of the present study.

The presence or absence of scintillation effects on trans-ionospheric radio signals have been extensively studied for electron density enhancements in the high-latitude ionosphere, but the effect of the steep plasma density gradients at the edge of depletions, such as polar holes are not as extensively studied. The purpose of this paper is to report on the effects of such steep density gradients on GNSS signals, observed in three multi-instrument case studies close to winter solstice.

## 2 Instrumentation

The European Incoherent Scatter Scientific Association (EISCAT) operates the EISCAT Svalbard Radar (ESR) at Longyearbyen (78.2° N, 16.0° E; 15.2° MLAT, 112.9° MLON) on Svalbard (Wannberg et al., 1997). The site consists of two antennas, a 32-meter parabolic dish and a 42-meter parabolic dish. The 42 m dish is fixed along the direction of the local geomagnetic field lines (azimuth -179°; elevation 81.6°), while the 32 m dish is steerable in both azimuth and



elevation. Observations of the electron density, electron temperature, ion temperature, and ion drift line of sight velocity in the ionosphere from this incoherent scatter radar (ISR) are used in this study.

The Super Dual Auroral Radar Network (SuperDARN) is a network of high latitude coherent scatter radars (Greenwald et al., 1995; Chisham et al., 2007; Nishitani et al,. 2019) that observe line-of-sight plasma velocities in the F-region. These measurements are assimilated using the map potential technique (Ruohoniemi and Baker, 1998), which

uses an ionospheric convection model to map the electrostatic potential pattern. Electrostatic equipotential lines are streamlines of ionospheric convection flows. As the plasma drift velocity is perpendicular to both the electric and magnetic fields in the F-region ($\underline{E} \times \underline{B}$ drift) the plasma convection pattern can be directly inferred from the electric potential maps.

GNSS signals detected by NovAtel GPStation-6 receivers at the Kjell Henriksen Observatory (KHO) (78.2° N,

16.0° E; 15.2º MLAT, 112.9º MLON ) can be used to infer the effects of the ionosphere on radio waves traveling though this medium. Amplitude scintillation is measured using the $S_4$ index, which is the square root of the variance of received power divided by the mean value of the received power (Briggs and Parkin, 1963). Phase scintillation is measured using the $\sigma_\phi$ index, which is the standard deviation of the detrended carrier phase $\phi$ in radians (Fremouw et al., 1978) over 60 seconds.

The IMF was observed by the Advanced Composition Explorer (ACE), which is a NASA Spacecraft orbiting the L1 Lagrangian point of the Earth Sun system, roughly 1.54 million km from the Earth (Zwickl et al., 1998). In addition to the x-, y- and z- components of the IMF the clock angle, given by $arctan \frac{|B_y|}{|B_z|}$, is also considered. When the clock angle is greater than 45 degrees either $|B_y|>|B_z|$ or $B_z<0$, in either case a two cell convection pattern is expected with antisunward flow drawing plasma from day to night across the polar cap (Thomas and Shepherd, 2018).

Total Electron Content (TEC) maps are used to put these measurements into context. These were obtained from the Madrigal Database at the MIT Haystack Observatory (Ridout and Coster, 2006; Vierinen et al., 2015). Two other indices are used within this study. The $K_p$ index is used as a proxy for disturbances to the geomagnetic field. The F10.7 cm solar flux is used as a proxy for solar activity. These indices were both obtained from the UK Solar System Data Centre (UKSSDC) at Rutherford Appleton Laboratory, UK.

## 3    Results

### 3.1    Case study: 17th December 2014

The 3-hourly $K_p$ values observed on 17[th] December 2014 between 12:00 and 23:59 UT ranged between 1- and 1+, indicating quiet conditions. The F10.7 cm solar flux was relatively high, the value of 198.5 sfu is typical of solar maximum. The IMF observed by the ACE spacecraft between 12:00 and 23:59 UT (Fig. 1) was characterised by a

positive value for the IMF $B_y$ (mean value 4.1 nT). IMF $B_z$ was more variable, but generally took smaller values (mean





value of 1.8 nT). The clock angle was generally greater than 45° until 19 UT, and the corresponding SuperDARN plots, a subset of which are shown in Fig. 5, show that a two cell convection pattern dominated until at least 20 UT.

Total Electron Content (TEC) maps (Fig. 2) show the overall plasma density throughout the high-latitude regions. The TEC maps at 12 UT and 15 UT show values of ~2 TECu (dark blue colour) in the polar cap. At 18 UT at 21 UT
larger electron densities can be observed crossing the polar cap in a two cell convection pattern, with values of ~15 TECu (yellow colour), indicating that plasma produced by photoionisation on the dayside is being drawn into the polar cap. This plasma is being drawn into the polar cap during relatively quiet conditions ($K_p$~1) and is consistent with a two cell convection pattern.

The electron densities and temperatures observed by the field-aligned 42 m dish of the EISCAT Svalbard Radar
(ESR) between 12:00 UT and 23:59 UT are shown in Fig. 3. A clear depletion in the electron densities is observed between approximately 16 and 18 UT at all altitudes. The electron and ion temperatures are not elevated at this time with values of approximately 1000 K, suggesting that this depletion is void of particle precipitation and did not arise from enhanced recombination due to Joule heating. In order to further investigate this depletion, a line plot of the maximum detected electron density from 90-400 km is shown (Fig. 4). In addition to the maximum density two other
values are present on the plot, the average value for the whole day, and 35% of the average value. The depletion was defined as when the electron density dropped below the 35% line and, in this case, the depletion was defined as starting at 16:29 UT and ending at 18:00 UT.

Fig. 5 shows the high-latitude convection pattern inferred from the SuperDARN radars for three representative times during the time that the electron density depletion was observed by the ESR. These clearly show a two cell convection
pattern, with plasma drawn antisunward across the polar cap. The ESR observes at a given location, which rotates under the convection pattern. The depletion identified in Fig. 4 is indicated by a black line. At midwinter Svalbard is in perpetual darkness. On 14th December the ground level terminator is at a maximum latitude of 68° N, which corresponds to a maximum magnetic latitude of 76° MLAT at 21 UT. This depletion is nightward of the terminator and the SuperDARN convection patterns suggest that this plasma is circulating in perpetual darkness. It is interpreted as a
polar hole.

The data collected by the GNSS receiver was from the GPS, Galileo and GLONASS systems and the receiver provides the azimuth and elevation of the satellite with respect to the receiver. This was converted into a latitude and longitude using the radio wave path and assuming that the data corresponds to 350 km in altitude, in line with previous studies (e.g. Cervera and Thomas, 2006; Forte and Radicella, 2002). At low elevation angles the GNSS TEC and
scintillation data can become unreliable due to multi-path issues, so observations at an elevation of less than 30° were discarded. This cut of has been used in previous studies, for example Mitchell et al. (2005). Signal lock times below 240 seconds were also discarded, in line with previous studies (e.g. van der Meeren et al., 2015). The satellite tracks were overlaid onto SuperDARN plots. (Fig. 5)



TEC and phase scintillation data from GNSS satellites were taken during times when the polar hole was observed. This hole is observed for 1.5 hours and several satellite paths are present during this time window. The GNSS TEC data clearly show lower TEC levels at and around the area marked by the ESR as a hole and, on some of the satellite trajectories, sharp changes can be seen with the edge of the hole. A one-to-one correspondence between the GNSS TEC data and the EISCAT data is neither expected or observed. It is highly likely that the polar hole will evolve during the time for which it is observed, and therefore the plots in figure 5 include both spatial and temporal variation. The ESR observes the polar hole for 91 minutes and the plasma velocity inferred from figure 5 at this location is of the order of150 m s$^{-1}$, indicating that the polar hole has a horizontal extent of some 800 km in a direction parallel to the plasma flow. In summary the combination of the EISCAT and GNSS TEC measurements indicate that the polar hole is present for an extended period of time (of the order of hours) over a large (hundreds of km) spatial scale.

Panels showing the location of phase scintillation on the satellite tracks are also shown in figure 5. A low threshold of 0.2 rad was used to identify phase scintillation. The purpose of this low threshold was to ensure that any possible indication of phase scintillation was included. Since TEC and scintillation are collected simultaneously, comparing the two might be expected to show increased scintillation where there are changes in TEC.  No scintillation was been seen on the edges of the holes.

### 3.2 Case study 2: 10th December 2015

The F10.7cm solar flux for this case was lower than in the first study, with a value of 108.5 sfu. The $K_p$ index was higher, with values of 3 from 12 to 18 UT and values of 4 at 21 and 24 UT, indicating an active state, but not storm levels. Once again the IMF was variable, with $B_z$ taking positive and negative values. $B_y$ was consistently larger than $B_z$ and dominated. As in the previous case study a two cell convection pattern was observed.

The TEC maps at 18 and 21 UT are shown in Fig. 7. As in the previous case study these indicate higher density plasma produced at lower latitudes being drawn across the polar cap within the high latitude convection pattern, with this effect maximising at 21 UT.

The 42 m ESR observations (Fig. 8) for this day show an electron density depletion that contains all the previously discussed markers. Using the same method as previously the hole was identified with the start and end position given as 15:15 and 16:43 UT. The 32 m ESR observations (Fig. 15) show a depletion at around 15 UT.

The high-latitude convection pattern was inferred from the SuperDARN radars (Fig. 11), with the location of the polar hole observed in the 42 m ESR observations, and GNSS TEC and phase scintillation measurements overlaid as in the previous case study. The 32 m ESR observations (Fig. 9) were directed poleward; indicating that this a polar hole rather than the ionospheric trough, which would be located equatorward of the radar. This second case study shows a more asymmetric convection pattern with a clear dominant dusk cell, drawing plasma across the polar cap from the pre-noon sector. The polar hole observed with the 42 m dish of the ESR was in the sunward return flow in the dusk convection cell.



The phase scintillation plot for 15:16 to 16:14 UT (upper right panel of Fig. 11) has some satellite trajectories passing through the hole boundary, but displays no significant scintillation on any of the paths. The later plot (second panel from the bottom on the right panel of Fig. 11) does contain phase scintillation seen however none of the elevated scintillation matches up to hole boundaries, instead, the scintillation is seen in regions of high and elevated electron density.

## 4   Discussion

A series of polar ionospheric holes have been detected in the high latitude nightside ionosphere in case studies close to winter solstice, under varying solar intensities and geomagnetic disturbance levels. The first study on 17[th] December 2014 saw high levels of solar activity (198.5 sfu) and quiet geomagnetic conditions. The second case study, on 10[th] December 2015 also had lower levels of solar activity of (108.5 sfu), but had more active geomagnetic conditions ($K_p$=3) than in the previous study. A third case study, under quiet geophysical conditions ($K_p \leq 2$) and moderate solar activity (F10.7 cm solar flux = 116.7 sfu) on 12[th] December 2015 showed similar results  (not shown).

Ionospheric polar holes contain much lower electron densities than those detected through the rest of the day, this study used the maximum density at a given time dropping 35% below the daily average maximum density to identify these holes. The changes in electron density are associated with large electron density gradients. Table 1 shows the electron density gradients and average hole electron density, based on observations from the ESR 42 m. The average polar hole density observed in this study is comparable to those previously reported of $10^8$-$10^{11}$ electrons·m$^{-3}$ (Obara and Oya, 1989, Benson and Grebowsky, 2001). Steep electron density gradients are observed at the edges of the holes, these are expressed in units of $\Delta N_e \cdot m^{-3} \cdot h^{-1}$. Although these gradients are expressed in units of h$^{-1}$ they were calculated from successive observations by the ESR 42 m (these measurements are typically one minute apart). The spatial extent of these holes was at least several hundred kilometres, as inferred from the GNSS TEC measurements (all studies) and the ESR 32 m observations (case study from 17[th] December 2014). Polar holes are usually associated with quiet geomagnetic conditions ($K_p$<2). It is notable that, on 10[th] December 2015, a polar hole was observed under more active geomagnetic conditions ($K_p$=3).

The IMF conditions during the time when the polar holes were observed, and for several hours beforehand, were appropriate for antisunward cross-polar convection. The ground level solar terminator for winter is only above 70° MLAT between 15 UT and slightly after 21 UT, reaching a maximum latitude of just under 76° MLAT on the dayside at around 21 UT, creating the possibility that plasma within the high-latitude convection pattern could circulate in perpetual darkness, thus undergoing recombination whilst simultaneously being insulated from photoionisation, or precipitation, creating a polar hole.

Phase scintillation has previously been observed to coincide with large plasma gradients such as on the edge of ionospheric enhancements such as polar cap patches (Jin et al., 2017), the tongue of ionisation (van der Meeren et al.,





2014), plasma structures associated with the aurora (Kinrade et al., 2013; Oksavik et al., 2015; van der Meeren et al.,
225    2015) and the mid-latitude trough (Pryse et al., 1991). The structures that cause scintillation arise due to the Gradient
Drift Instability and/or the Kelvin Helmholtz Instability (Keskinen and Ossakow, 1983; Carlson et al., 2008). In the
present study, once the boundaries and the large electron density gradients associated with them were identified these
boundaries were investigated for elevated levels of phase scintillation. A threshold of 0.2 rad was used, the purpose of
this low value was to ensure that any possible indication of phase scintillation was included. Across all of the observed
230    GNSS points coinciding with the polar hole boundaries no such levels of phase scintillation were detected. Phase
scintillation usually dominates at high latitude (e.g., Prikryl et al., 2015), although amplitude scintillation has also been
observed (e.g. Mitchell et al., 2005). The present study focuses upon phase scintillation as no amplitude scintillation,
defined as when the S4 index was greater than 0.2, was observed on any of the TEC gradients at the boundaries of the
polar holes.

235        This is not the first time a plasma density enhancement has been observed without corresponding phase scintillation.
Van der Meeren et al. (2016) observed a Sun-aligned polar cap arc under quiet geomagnetic conditions without
corresponding scintillation.  In the present study some phase scintillation was observed, however, these points coincide
with increases in TEC and the edges of spikes in electron densities at other locations. In the second case study (10[th]
December 2015) phase scintillation was observed at a point associated with elevated TEC (lower right panels of Fig.
240    11), but this was not associated with the assumed boundary of the polar hole.

When phase scintillation was observed it was always associated with electron density gradients, but converse is not
always true. Therefore it appears that some minimum level of overall electron density is needed for phase scintillation
to occur. Given that it is the presence of small scale structures that cause scintillation, this suggests that these small
scale structures have not arisen.

245        Figure 12 shows phase scintillation as a function of TEC and TEC rate of change. This figure also includes data
from a third study, using data from 12[th] December 2015, which was consistent with the interpretation presented here,
but which has been omitted in the interest of concision. Low scintillation can be seen at all TEC levels and for a majority
of the range of TEC rates of change. On the other hand, elevated scintillation levels are only seen above approximately
6 TECU suggesting that a minimum electron density is required.  This is not a new idea, in his review paper Aarons
(1982) commented 'if the ionosphere is perturbed on a percentage basis, *change in N* in the trough will be small since
*N* is low; scintillations will then be low.' The current paper provides observational evidence to support this suggestion
that a minimum electron density is required. The current paper is also consistent with suggestions made by Prikryl et
al. (2015), where the strongest phase scintillations were found to be highly collocated with regions that are ionospheric
signatures of the coupling between the solar wind and magnetosphere. Polar holes appear to be areas of weak coupling,
hence less scintillation.

Further developments upon this work would expand the GNSS coverage of the polar holes discussed to a larger
number of examples under a wider range of geophysical conditions. A further development would be to track the





evolution of polar ionospheric holes by making observations with a higher temporal resolution at a large number of

regularly spaced locations. The advent of EISCAT-3D (McCrea et al., 2015), which will give unprecedented temporal

and spatial coverage, will enable such studies in the European sector of the high-latitude ionosphere. The ability to

observe the evolution of polar holes over time will give a new, deeper, understanding of these features and how they

influence practical radio systems such as GNSS.

## 5   Conclusions

Polar ionospheric holes are regions of electron density depletions containing large electron density gradients at their

boundaries. These holes were observed during geomagnetically quiet and moderately disturbed conditions and under a

range of solar activities using the EISCAT Svalbard Radar (ESR) and measurements from GNSS satellites.  Steep

electron density gradients have been associated with phase scintillation at GNSS frequencies in previous studies,

however no enhanced scintillation was detected upon the electron density gradients at these boundaries. Phase

scintillation was only observed when electron density levels were elevated above 6 TECU and a gradient was present

implying that both a minimum electron density level and a sharp gradient in the election density must be present for

instability mechanisms to produce scintillation structures.

## Author contribution

This work was led by Luke Jenner, under the guidance of Alan Wood. Kjellmar Oksavik provided the GNSS TEC

and scintillation data, together with guidance regarding their interpretation. Tim Yeoman and Alexandra Fogg provided

the SuperDARN electric potential maps, together with guidance regarding their interpretation. Anthea Coster provided

the TEC maps, together with guidance regarding their interpretation. All authors contributed to the discussion. The

manuscript was prepared by Luke Jenner and Alan Wood.

## Competing interests

The authors declare that they have no conflict of interest.

## Acknowledgements


EISCAT is an international facility supported by the national science councils of China, Finland, Japan, Norway,

Sweden, and the United Kingdom. The assistance of Ingemar Häggström and colleagues at the EISCAT Scientific

Association in running the experiments is gratefully acknowledged. The data used in this paper is publicly available at

https://www.eiscat.se. The assistance of Steve Crothers and Matthew Wild at Rutherford Appleton Laboratory with the





data processing is gratefully acknowledged. The GNSS TEC and scintillation data were provided by Kjellmar Oksavik at the University of Bergen, and is supported by the Norwegian Research Council under contracts 212014 and 223252. The authors acknowledge the use of SuperDARN data, data for which is available at https://vt.superdarn.org. SuperDARN is a collection of radars funded by national scientific funding agencies of Australia, Canada, China, France, Italy, Japan, Norway, South Africa, United Kingdom and the United States of America.' Alexandra Fogg is supported

by a studentship from the Science and Technology Facilities Council (UK). The assistance of Nathan Brown with the production of Fig. 5 and Fig. 11 is gratefully acknowledged. GPS TEC data products and access through the Madrigal distributed data system are provided to the community (http://www.openmadrigal.org) by the Massachusetts Institute of Technology (MIT) under support from US National Science Foundation grant AGS-1242204.  Data for TEC processing is provided from the following organizations: UNAVCO, Scripps Orbit and Permanent Array Center, Institut

Geographique National, France, International GNSS Service, The Crustal Dynamics Data Information System (CDDIS), National Geodetic Survey, Instituto Brasileiro de Geografia e Estatística, RAMSAC CORS of Instituto Geográfico Nacional de la República Argentina, Arecibo Observatory, Low-Latitude Ionospheric Sensor Network (LISN), Topcon Positioning Systems, Inc., Canadian High Arctic Ionospheric Network, Centro di Ricerche Sismologiche, Système d'Observation du Niveau des Eaux Littorales (SONEL), RENAG : REseau NAtional GPS

permanent, GeoNet - the official source of geological hazard information for New Zealand, GNSS Reference Networks, Finnish Meteorological Institute, and SWEPOS - Sweden. Access to these data is provided by madrigal network via: http://cedar.openmadrigal.org/.The $K_p$ index and F10.7 cm solar flux were obtained from the UK Solar System Data Centre at Rutherford Appleton Laboratory. These can be accessed at https://www.ukssdc.ac.uk/. The IMF data were provided by N. Ness and obtained from the CDAWeb at https://cdaweb.gsfc.nasa.gov/.

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



**Figures**



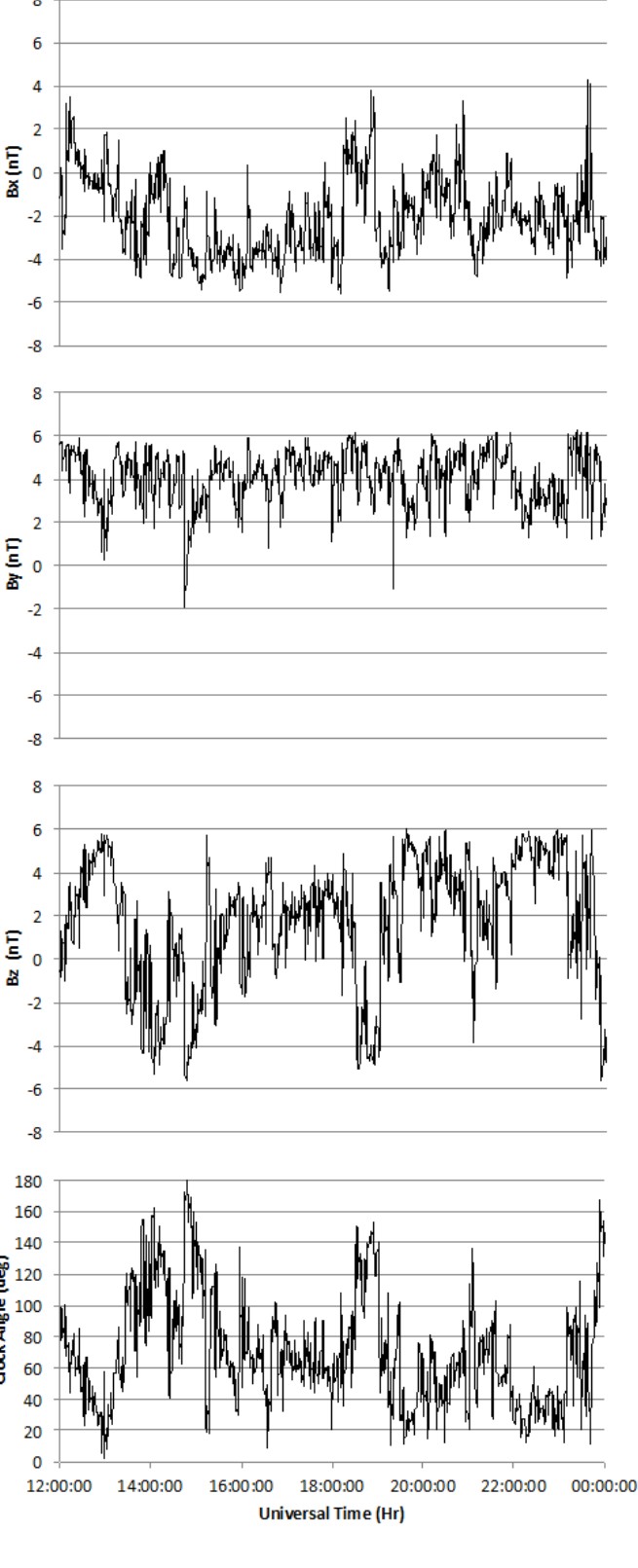





**Fig. 1.** The x-, y- and z-components of the IMF, and the clock angle observed by the ACE spacecraft between 12:00 UT and 23:59 UT on 17<sup>th</sup> December 2014. The data has not been time shifted to allow for the transit time from ACE to the Earth, which was approximately 73 minutes.

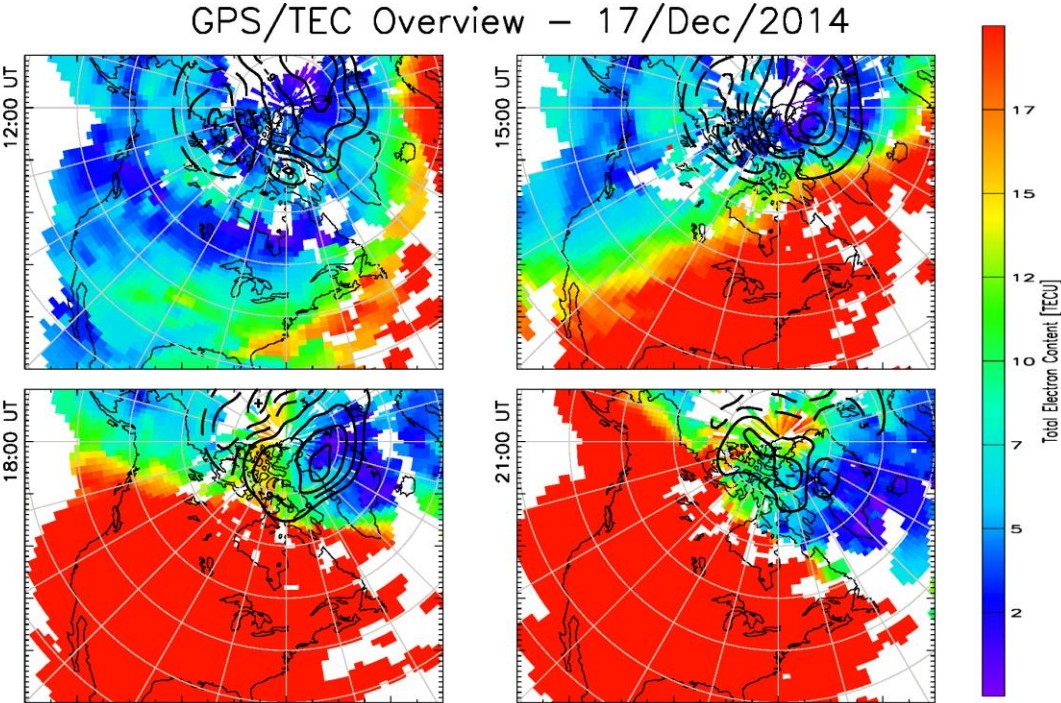

**Fig. 2.** TEC maps for the 17<sup>th</sup> December 2014 extrapolated from TEC collected by a network of GNSS receivers at three hourly intervals between 12 UT and 21 UT.



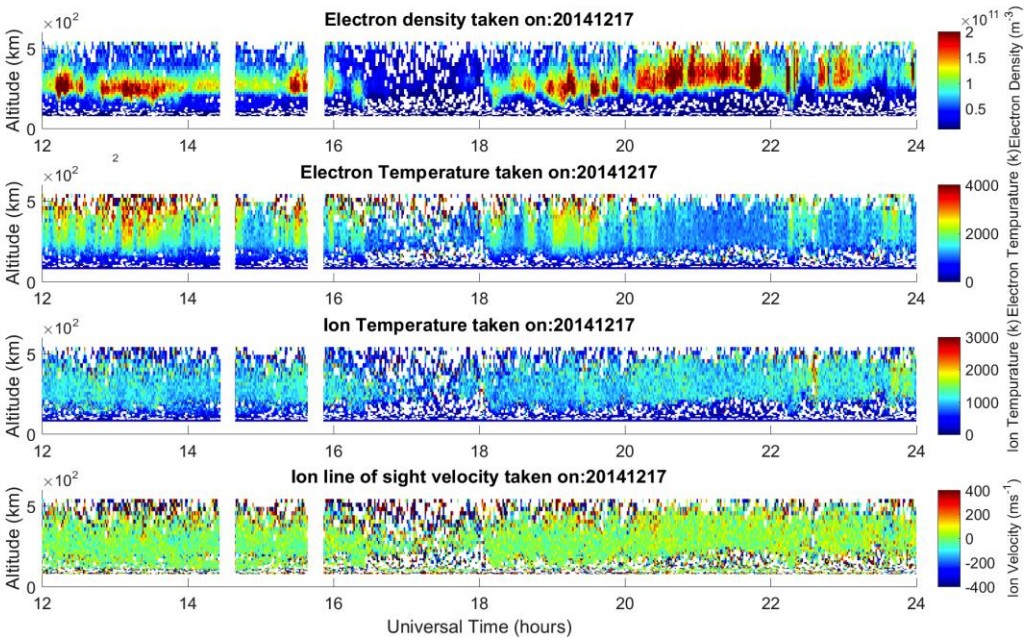

**Fig. 3.** Electron densities, electron temperatures, ion temperatures, and ion drift line of sight velocity measured by the 42 m dish of the ESR observing at an azimuth of 184.5∘ and an elevation of 81.6∘ between 12:00 UT and 23:59 UT on 17th December 2014.



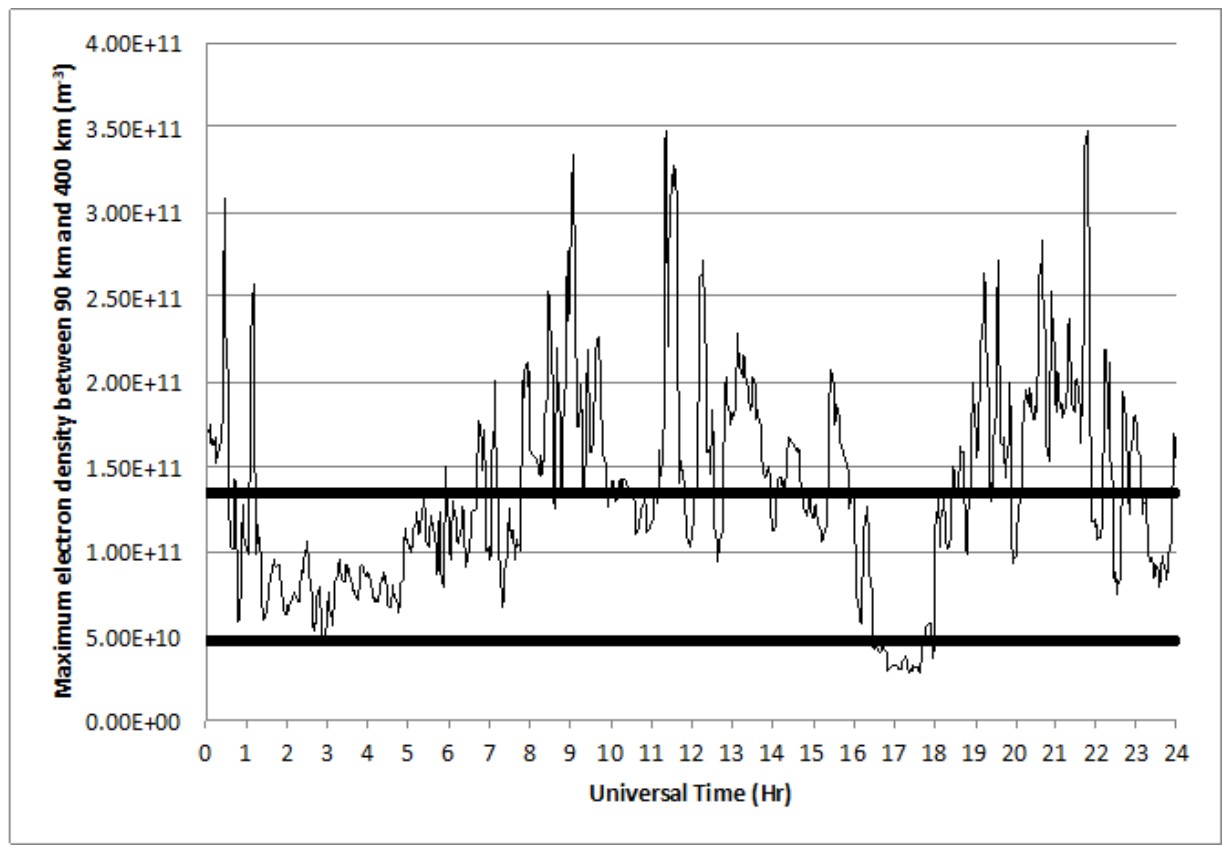

**Fig. 4.** Maximum electron density between 90 and 400 km for ESR 42 m observation on the 17[th] December 2014 at one minute resolution.
A five point running mean was applied to these data. The upper horizontal line is the average value and the lower horizontal line is 35% of
the average. A hole can be seen between 16:29 and 18:00 UT.



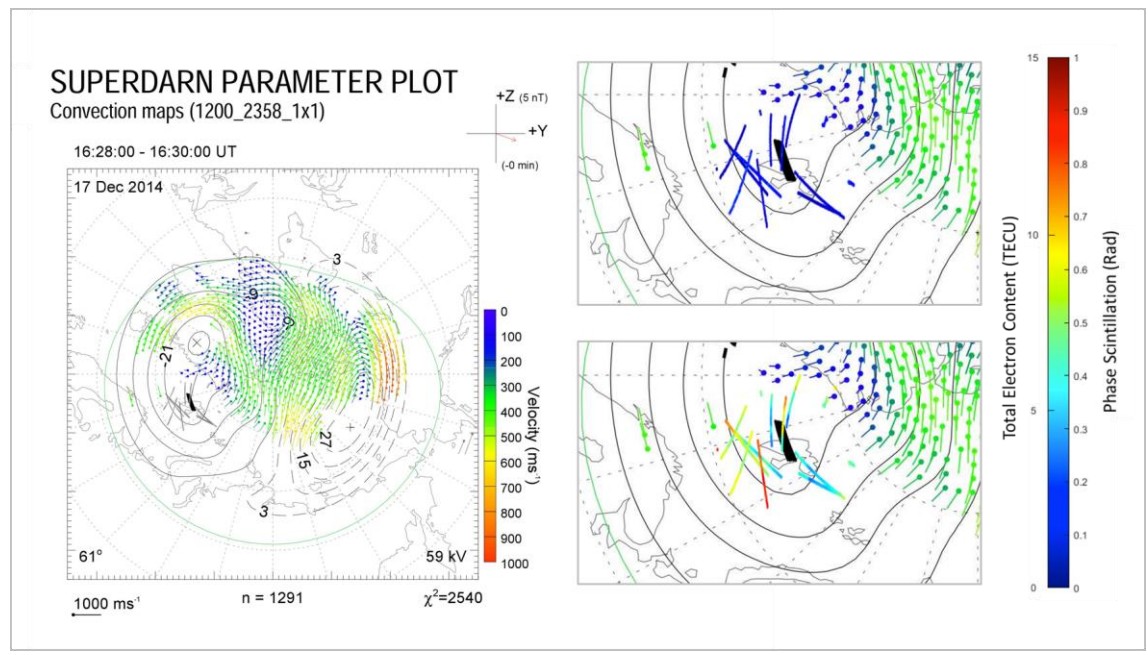

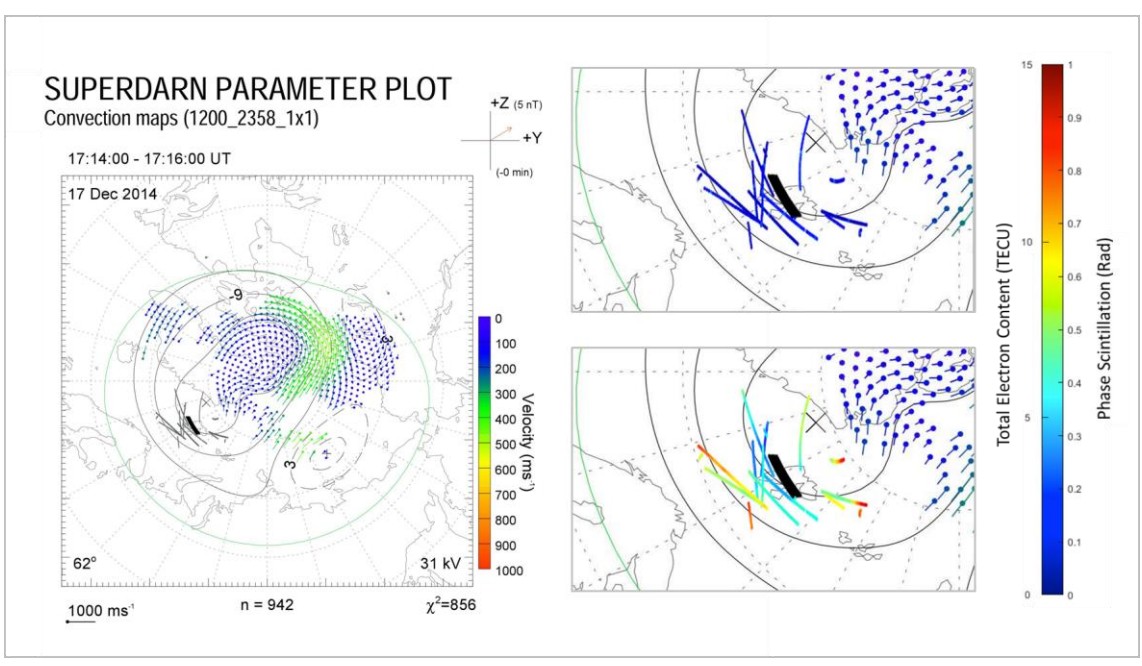



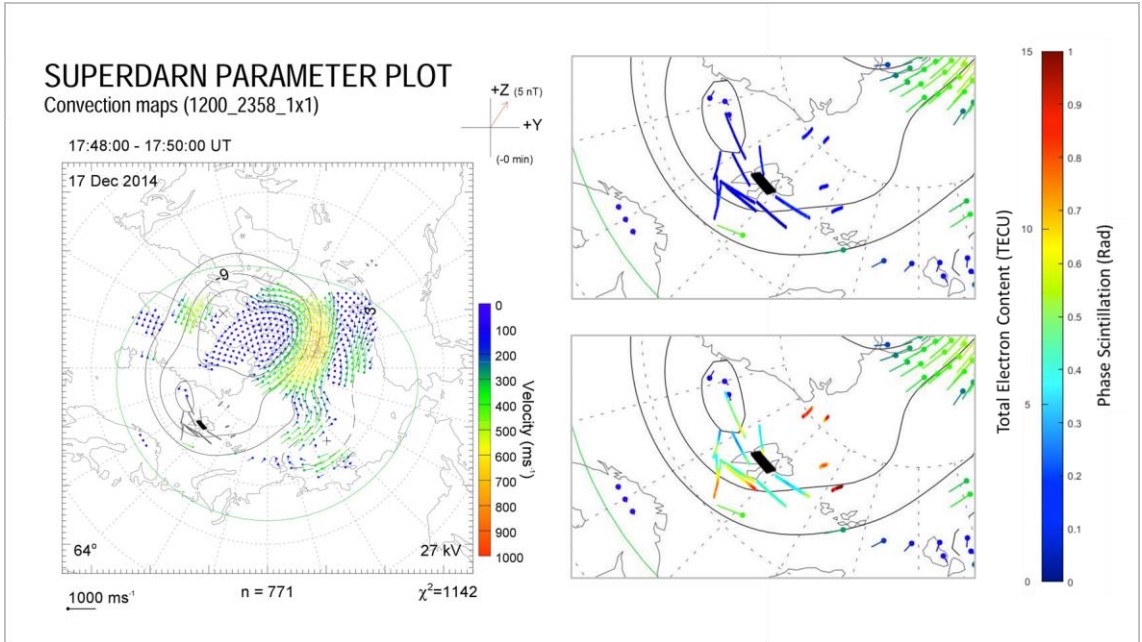

**Fig. 5.** Electric potential patterns inferred from the SuperDARN radars for 16:28 UT, 17:14 UT, and 17:48 UT on 17[th] December 2014 as a function of geomagnetic latitude and magnetic local time. Magnetic noon is at the top of each plot with dusk and dawn on the left- and right-hand sides respectively. Magnetic latitude is indicated by the grey dashed circular lines in 10.0◦ increments. The grey lines show the location of satellite passes from GNSS satellites, assuming an ionospheric intersection of 350 km. The SuperDARN plot from 16:28 UT includes satellite passes from 16:00-16:58 UT, the 17:14 UT plot includes satellite passes from 16:58-17:28 UT, and the 17:48 UT plot includes satellite passes from 17:28-18:02 UT. These time intervals were chosen as inspection of the whole SuperDARN data set at two minute resolution indicated that the convection patterns were relatively stable during these intervals. The right hand side of the panels show the area around the satellite passes in more detail. The multi-coloured colours represent phase scintillation (upper panel in each pair) and TEC (lower panel in each pair). The thick black line indicates the position of the polar hole observed with the 42 m dish of the EISCAT Svalbard Radar.



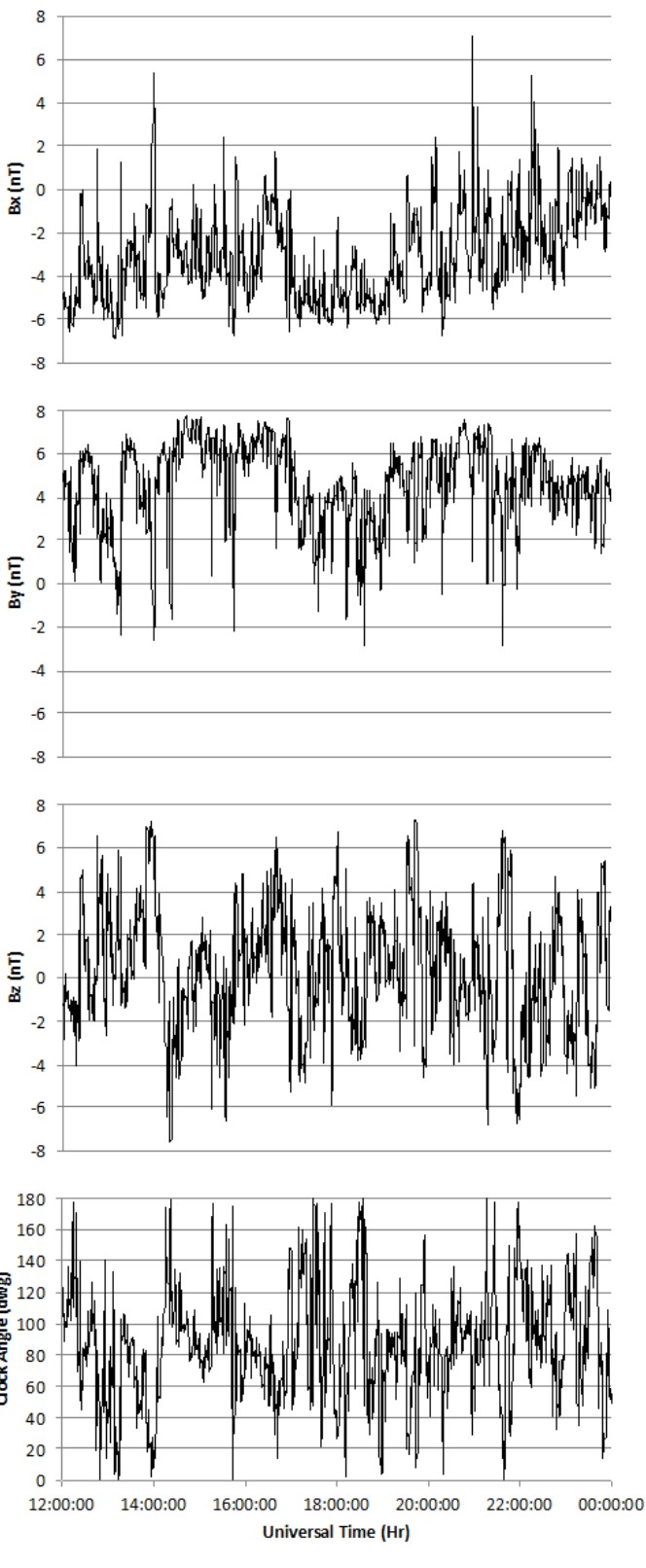



**Fig. 6.** The x-, y- and z-components of the IMF, and the clock angle observed by the ACE spacecraft between 12:00 and 23:59 UT on 10$^{th}$
December 2015, in the same format as Fig. 1. The data has not been time shifted to allow for the transit time from ACE to the Earth, which
was approximately 45 minutes.

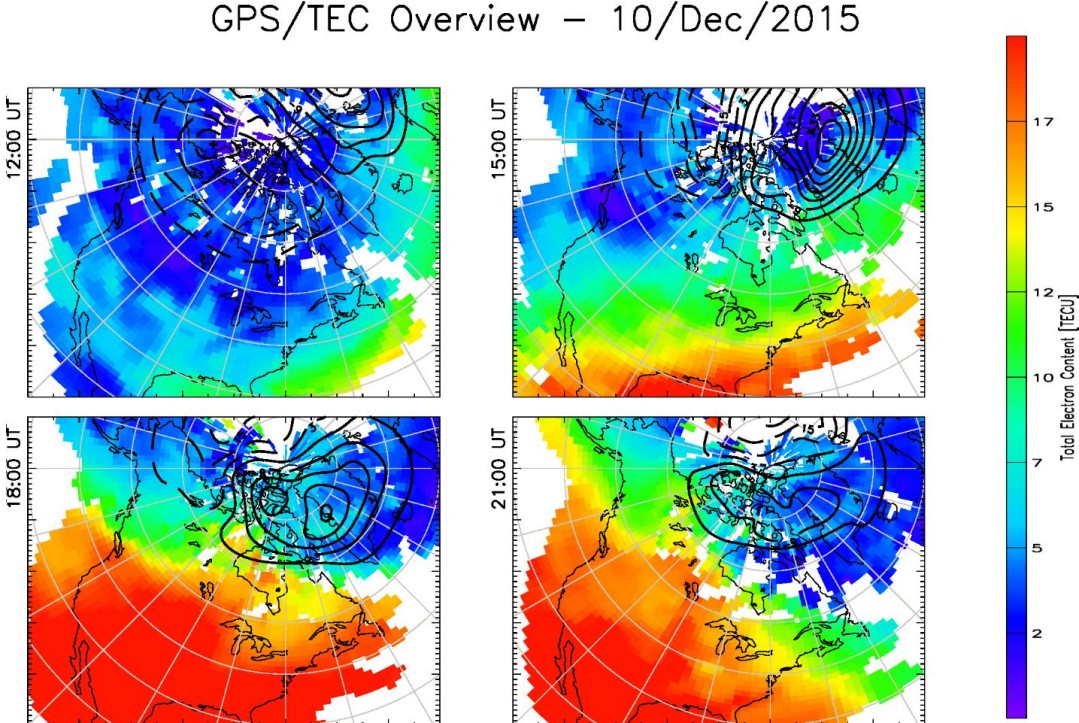

**Fig. 7.** TEC maps for the 10$^{th}$ December 2015 extrapolated from TEC collected by a network of GNSS receivers at three hourly intervals
between 12 and 21 UT.





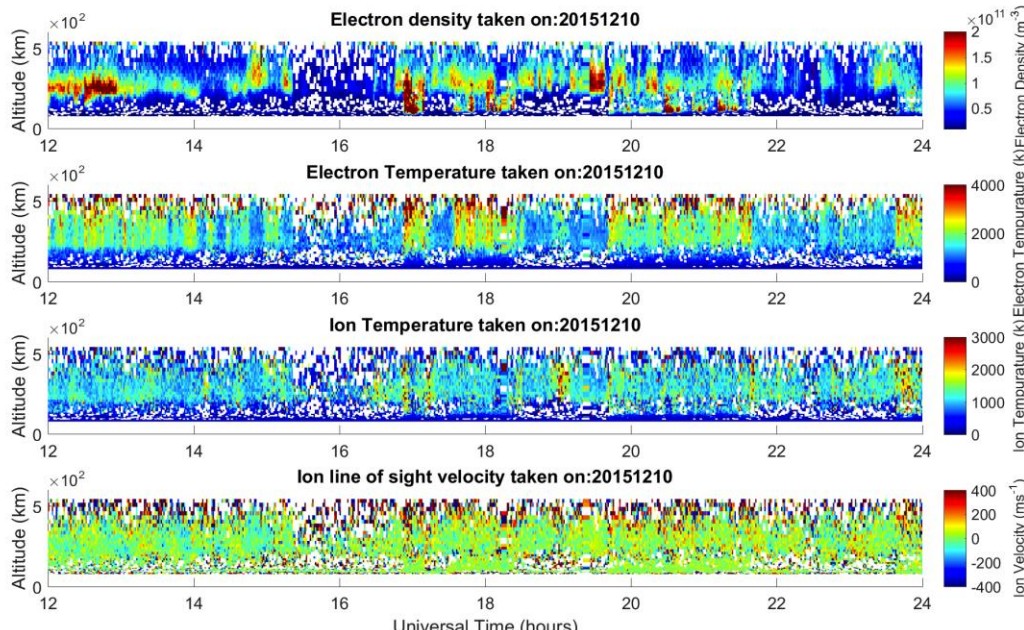


**Fig. 8.** Electron densities, electron temperatures, ion temperatures, and ion drift line of sight velocity measured by the 42 m dish of the ESR observing at an azimuth of 184.5◦ and an elevation of 81.6◦ between 12:00 and 23:59 UT on 10th December 2015.





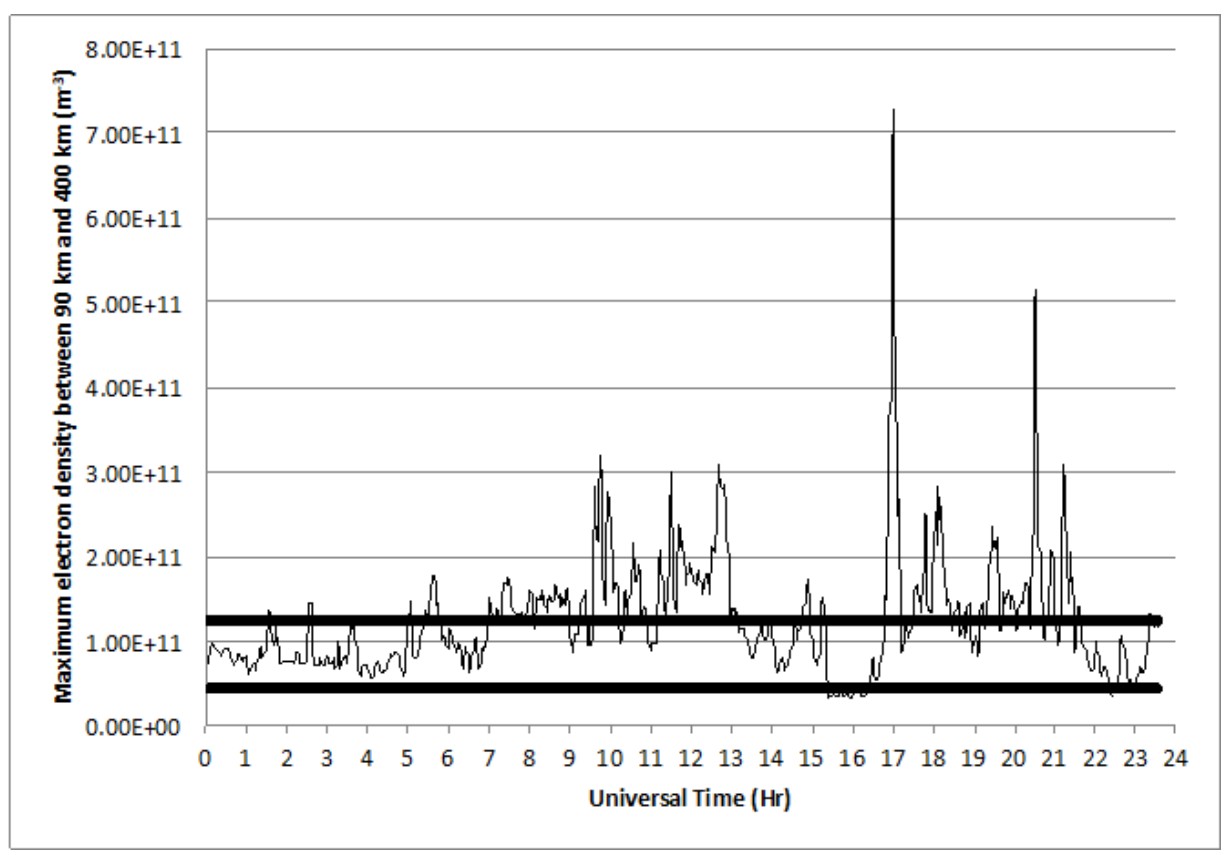

**Fig. 9.** As Fig. 4 but for 10th December 2015. A polar hole can be seen between 15:24 and 16:25 UT.

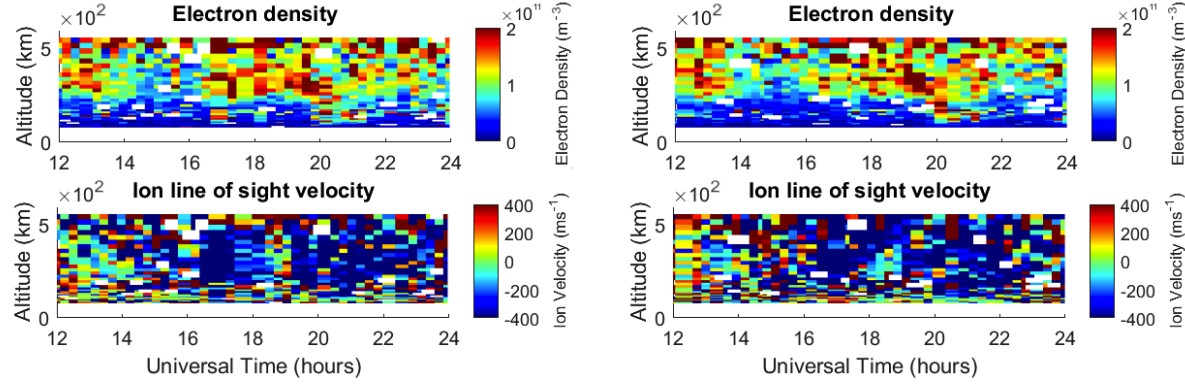


**Fig. 10.** Electron densities and ion drift line of sight velocities observed by the 32 m dish of the ESR at -43° azimuth and 30° elevation (left hand side) and at -14° azimuth and 30° elevation (right hand side) between 12:00 and 23:59 UT on 10th December 2015.

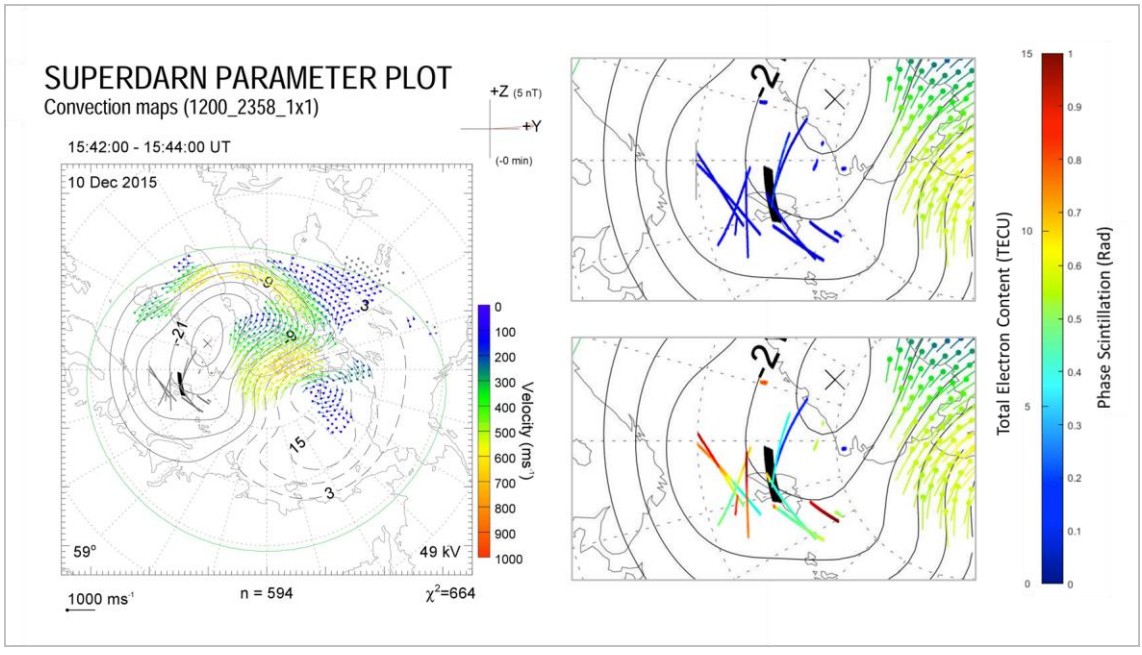

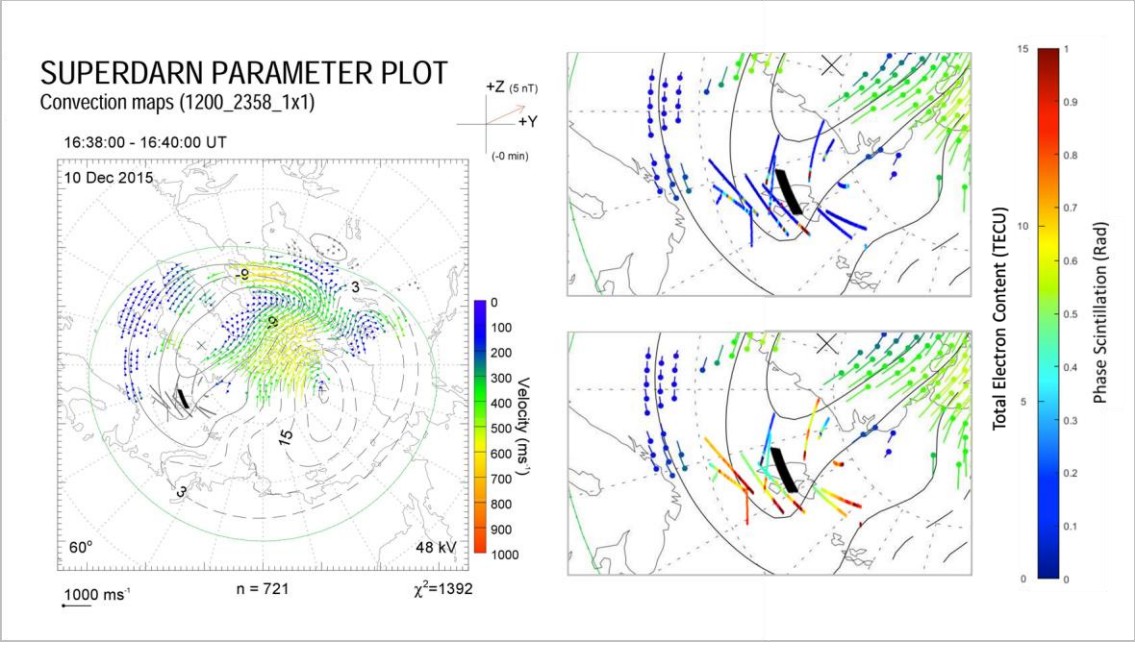

**Fig. 11.** Electric potential patterns inferred from the SuperDARN radars for 15:42 UT and 16:38 UT on 10[th] December 2015, with data from GNSS satellites overlaid in the same format as Fig. 5. The intervals for which the satellite passes were plotted are from 15:16-16:14 UT (15:42 UT plot) and from 16:14-17:04 UT (16:38 UT plot).





| Date | 1st Edge $\Delta N_e \cdot m^{-3} \cdot h^{-1}$ | 2nd Edge $\Delta N_e \cdot m^{-3} \cdot h^{-1}$ | Average Hole $N_e \cdot m^{-3}$ |
|---|---|---|---|
| 17/12/2014 | 1.0E+11 | 9.1E+10 | 4.0E+10 |
| 10/12/2015 | 3.5E+11 | 1.6E+11 | 2.2E+10 |
| 12/12/2015 | 7.9E+10 | 1.0E+11 | 1.8E+10 |

**Table 1** – The electron density gradient at each edge of the polar hole and the average electron density inside the hole at 350 km observed by ESR 42 m.