# Peer review of "Plasma density gradients at the edge of polar ionospheric holes: The absence of phase scintillation"

_Annales Geophysicae, 2019_

## Referee Comment (RC1) · Anonymous Referee #1 · 24 Sep 2019

Comments on Plasma density gradients at the edge of polar ionospheric holes: The presence and absence of phase scintillation By Jenner et al.

The paper presents a study about 3 cases (2 reported) of polar holes and discusses their role in acting as scintillation driver for GNSS signals. The paper is clear and well written and worth to be published after minor revisions.

Major issues. • The authors seem to ignore all the recent work made to better clarify the meaning and definition of phase scintillation at high latitude. In particular, it has been recently addressed that is more appropriate to distinguish between "phase fluctuations", due to refractive effects on the signals and of deterministic nature, and "phase

scintillation", due to diffractive effects, of stochastic nature and ruled by the Fresnel's filtering mechanism. This follows also discussion about the not appropriateness of 0.1 Hz as fixed cutoff frequency for phase dentrending in SigmaPhi calculation. This has been introduced in the early 2000 in the works by Forte (2002, 2005), by Forte and Radicella (2002) and Beach (2005). Then, in the last decade, it has been deeply investigated and discussed on case studies (Mushini et al., 2012; Wang et al., 2018; McCaffrey and Jayachandran, 2019) and at a statistical level (De Franceschi et al., 2019). The authors are encouraged to revise the text and results to take into consideration the recent achievements in that sense. • The title of the paper is misleading, as more emphasis is given to the fact that polar holes doesn't results into meaningful phase fluctuations. • In addition, my personal guess is that the effect of the polar holes can be seen in the scintillation indices if they are calculated by considering not 1-minute value, but lower time windows (say 10 seconds). Of some inspiration can be the work by Smith et al. (2008).

Minor issues • In the abstract I would "measurements from GNSS receivers" and not "measurements from GNSS satellites" • Figure 5 is cited before figure 2,3 and 4. • The IMF data provided by ACE and available in CDAWEB are also provided time-shifted to the Nose of the Earth's Bow Shock. I encourage to report this. • Figure 3, bottom panel. The adopted color code is not suitable to catch positive/negative variations. I would suggest to use a blue/white/red for a better reading of the figure. • The threshold 0.2 radians for SigmaPhi is commonly used (0.25/0.3 too) as able to identify moderate to strong scintillation regimes. I would suggest to cite this.

References: Beach, T. L. Perils of the GPS phase scintillation index ($\sigma\varphi$). Radio science, 41(5) (2006). De Franceschi, G., Spogli, L., Alfonsi, L., Romano, V., Cesaroni, C., & Hunstad, I. (2019). The ionospheric irregularities climatology over Svalbard from solar cycle 23. Scientific reports, 9. Forte, B. & Radicella, S. M. Problems in data treatment for ionospheric scintillation measurements. Radio Science, 37(6) (2002). Forte, B. et al. Identification of scintillation signatures on GPS signals originating from plasma structures detected with EISCAT incoherent scatter radar along the same line of sight. J. Geophys. Res. Space Physics 122, 916–931, https://doi.org/10.1002/2016JA023271 (2017). Forte, B. On the relationship between the geometrical control of scintillation indices and the data detrending problems observed at high latitudes. Annals of Geophysics, 50(6) (2002). Forte, B. Optimum detrending of raw GPS data for scintillation measurements at auroral latitudes. J. Atmos. Sol. Terr. Phys. 67, https://doi.org/10.1016/j.jastp.2005.01.011 (2005). McCaffrey, A. M. & Jayachandran, P. T. Determination of the Refractive Contribution to GPS Phase "Scintillation". Journal of Geophysical Research: Space Physics (2019). Mushini, S. C., Jayachandran, P. T., Langley, R. B., MacDougall, J. W. & Pokhotelov, D. Improved amplitude-and phase-scintillation indices derived from wavelet detrended high-latitude GPS data. GPS solutions 16(3), 363–373 (2012). Smith, A. M., Mitchell, C. N., Watson, R. J., Meggs, R. W., Kintner, P. M., Kauristie, K., & Honary, F. (2008). GPS scintillation in the high arctic associated with an auroral arc. Space Weather, 6(3). Wang, Y. et al. Experimental evidence on the dependence of the standard GPS phase scintillation index on the ionospheric plasma drift around noon sector of the polar ionosphere. Journal of Geophysical Research: Space Physics 123(3), 2370–2378 (2018).
* * *

---

## Short Comment (SC1) · 13 Nov 2019

**Response to reviewer 1 of Jenner et al., Plasma density gradients at the edge of polar ionospheric holes: The absence of phase scintillation**

We thank reviewer 1 for their detailed and careful review of this paper. We note that reviewer 1 stated that this paper is clear, well written and worthy of publication after minor revisions. We thank reviewer 1 for this comment.

Reviewer 1 has raised a number of important points. All of these have been addressed below, and we believe that these substantially enhance the manuscript.

**Comment 1: The meaning and definition of phase scintillation at high-latitudes**

The reviewer commented that this was missing from the introduction. We had omitted this in the interests of concision, but the review is absolutely right that it needs to be included. A brief but comprehensive summary is now given in lines 117-130 using many of the references suggested by the reviewer.

**Comment 2: Title of the paper**

We agree that the suggested modification gives a better title for this paper, and have amended the title.

**Comment 3: Would phase scintillation be observed at lower time windows?**

This is an excellent suggestion, and one worthy of investigation. The effect of the length of the sampling window on phase scintillation indices would be a fascinating next step, and one which shall certainly be investigated in future experiments. A comment to this effect has been added in lines 425-429. As only the scintillation indices (and not the raw 50 Hz data) have been archived it is not possible to investigate within this study, and we have used the 60 second window which is commonly used within the community.

**Comment 4: Abstract should refer to GNSS receivers, rather than GNSS satellites**

Agreed. This change has been made (line 20). Similar changes have been made to lines 263 and 442.

**Comment 5: Figure 5 is cited before figures 2, 3 and 4.**

Thank you for pointing this out. Line 182 has been re-phrased accordingly.

**Comment 6: ACE data is available time-shifted to the bow shock**

Indeed it is. We have re-produced figures 1 & 6 and modified the text accordingly.

**Comment 7: EISCAT data (figures 3, 8 and 10): Choice of colour scale and scaling of velocity data**

The reviewer suggested changing the colour scale on these plots from red-green-blue to red-white-blue. We have retained the red-green-blue colour scale, as this is commonly used for EISCAT data and so enables an easy comparison to other data sets in the published literature.

The reviewer also commented that, for Fig. 3 (bottom panel) "The adopted color code is not suitable to catch positive/negative variations." This is true, but the choice was deliberate. The same scale was used for the velocity information in figures 3, 8 and 10. This was so that the difference between the larger velocities in Fig. 10 for the radar observing at low elevation and the smaller (negligible) velocities

at high elevation (figures 3 and 8) could be clearly seen.  However, we did not make this clear in the paper, and thank the reviewer for their comment. We have amended the text in lines 205-208, 274-275, 307-308 and 324-329 accordingly.

**Comment 8: Citation of chosen threshold for SigmaPhi**

A discussion of different thresholds (along with suitable citations) has been added to lines 279-282.

---

## Short Comment (SC2) · 13 Nov 2019

Revisions have been made to this manuscript, on the basis of the insightful comments made by reviewer RC1. Two versions of the revised manuscript has been uploaded. Jenner_et_al_revised.pdf includes all of the revisions, with major changes highlighted. Jenner_et_al_track_changes.pdf includes details of every individual change.

Please also note the supplement to this comment:
https://www.ann-geophys-discuss.net/angeo-2019-112/angeo-2019-112-SC2-supplement.pdf

**Supplement:**

[revised manuscript text omitted]

---

## Short Comment (SC3) · 13 Nov 2019

Revisions have been made to this manuscript, on the basis of the insightful comments made by reviewer RC1. Two versions of the revised manuscript has been uploaded. Jenner_et_al_revised.pdf includes all of the revisions, with major changes highlighted. Jenner_et_al_track_changes.pdf includes details of every individual change.

Please also note the supplement to this comment: https://www.ann-geophys-discuss.net/angeo-2019-112/angeo-2019-112-SC3-supplement.pdf

[Figure]

**Supplement:**

[revised manuscript text omitted]

---

## Referee Comment (RC2) · Anonymous Referee #2 · 16 Nov 2019

This manuscript presents new research about polar holes and associated electron density gradients and scintillations. It is well structured and well written. It has a clear objective and qualitative results which are presented in a well understandable way. I recommend its publication after minor comments are addressed.

1. The conclusions are currently described in a rather vague way. In the abstract, the last sentence "It may be that . . ." should be revised. In the conclusions section, also the last sentence should be revised. Currently it reads more like a summary instead of a conclusion. I recommend splitting the last sentence after ". . . gradient was present" and rephrase the second part of the sentence to a

conclusion.

2. It seems that the objective of the paper is to present observational proof for the comment in Aarons (1982) as described in Lines 251-253. I recommend to add this to the last part of the introduction.

3. Line 166 "of150 m s-1": insert space

4. Line 170 "was been": revision needed

5. Line 176 "values of": revision needed

6. Line 200 replace "saw" by something more intuitive like e.g. "was characterized by "

7. Line 227 insert a comma after "identified"

8. Table 1: use same exponent in each cell to make the numbers better comparable.

---

## Author Comment (AC1) · 8 Dec 2019

Response to reviewer 1 of Jenner et al., Plasma density gradients at the edge of polar ionospheric holes: The absence of phase scintillation

*** This document includes details of all responses to reviewer 1. The same information is contained in the document which gives a response to both reviewers ***

We thank the reviewer for their detailed and careful review of this paper. We note that the reviewer stated that this paper is clear, well written and worthy of publication after minor revisions. We thank the reviewer for this comment.

[Figure]

Reviewer 1 raised a number of important points. All of these have been addressed below, and a revised version of the manuscript has been submitted. For the convenience of the reviewer, two versions have been uploaded. 'Jenner_et_al_track_changes' shows every individual change and 'Jenner_et_al_revised' has all changes implemented, with major changes highlighted in yellow. We believe that these revisions substantially enhance the manuscript.

Detailed response to comments from Anonymous Reviewer 1

Comment 1: The meaning and definition of phase scintillation at high-latitudes

The reviewer commented that this was missing from the introduction. We had omitted this in the interests of concision, but the review is absolutely correct that it needs to be included. A brief but comprehensive summary is now given in lines 114-128 using many of the references suggested by the reviewer.

Comment 2: Title of the paper

We agree that the suggested modification gives a better title for this paper, and have amended the title.

Comment 3: Would phase scintillation be observed at lower time windows?

This is an excellent suggestion, and one worthy of investigation. The effect of the length of the sampling window on phase scintillation indices would be a fascinating next step, and one which shall certainly be investigated in future experiments. A comment to this effect has been added in lines 422429. As only the scintillation indices (and not the raw 50 Hz data) have been archived it is not possible to investigate within this study, and we have used the 60 second window which is commonly used within the community.

Comment 4: Abstract should refer to GNSS receivers, rather than GNSS satellites

Agreed. This change has been made (line 19). Similar changes have been made to lines 261 and 438.

Comment 5: Figure 5 is cited before figures 2, 3 and 4.

Thank you for pointing this out. Line 181 has been re-phrased accordingly.

Comment 6: ACE data is available time-shifted to the bow shock

Indeed it is. We have re-produced figures 1 & 6 and modified the associated text (lines 175-182 and lines 284-288) accordingly.

Comment 7: EISCAT data (figures 3, 8 and 10): Choice of colour scale and scaling of velocity data

The reviewer suggested changing the colour scale on these plots from red-green-blue to red-whiteblue. We have retained the red-green-blue colour scale, as this is commonly used for EISCAT data and so enables an easy comparison to other data sets in the published literature. The reviewer also commented that, for Fig. 3 (bottom panel) "The adopted color code is not suitable to catch positive/negative variations." This is true, but the choice was deliberate. The same scale was used for the velocity information in figures 3, 8 and 10. This was so that the difference between the larger velocities in Fig. 10 for the radar observing at low elevation and the smaller (negligible) velocities at high elevation (figures 3 and 8) could be clearly seen. However, we did not make this clear in the paper, and thank the reviewer for their comment. We have amended the text in lines 203-208 and 321-325 accordingly.

Comment 8: Citation of chosen threshold for SigmaPhi

A discussion of different thresholds (along with suitable citations) has been added to lines 276-280.

Please also note the supplement to this comment:
https://www.ann-geophys-discuss.net/angeo-2019-112/angeo-2019-112-AC1-supplement.pdf

[Figure]

**Supplement:**

[revised manuscript text omitted]

---

## Author Comment (AC2) · 8 Dec 2019

Response to reviewer 2 of Jenner et al., Plasma density gradients at the edge of polar ionospheric holes: The absence of phase scintillation

*** This document includes details of all responses to reviewer 2. The same information is contained in the document which gives a response to both reviewers ***

We thank the reviewer for their detailed and careful review of this paper. We note that the reviewer stated that this paper is clear, well written and worthy of publication after minor revisions. We thank the reviewer for this comment.

[Figure]

Reviewer 2 raised a number of important points. All of these have been addressed below, and a revised version of the manuscript has been submitted. For the convenience of the reviewer, two versions have been uploaded. 'Jenner_et_al_track_changes' shows every individual change and 'Jenner_et_al_revised' has all changes implemented, with major changes highlighted in yellow. We believe that these revisions substantially enhance the manuscript.

Detailed response to comments from Anonymous Reviewer 2

Comment 1: Purpose of the paper as discussed in the abstract and conclusion

The reviewer commented that "The conclusions are currently described in a rather vague way. In the abstract, the last sentence "It may be that . . ." should be revised. In the conclusions section, also the last sentence should be revised." We agree with this comment that the conclusions should be stronger, and have revised lines 441-446 accordingly. We also amended lines 435-439 to enhance the clarity of the paper. A corresponding change has been made in the abstract to line 23.

Comment 2: Purpose of the paper as discussed in the introduction

The reviewer commented that "It seems that the objective of the paper is to present observational proof for the comment in Aarons (1982) as described in Lines 251-253. I recommend to add this to the last part of the introduction." We agree with this comment that this should be more explicit, and have revised lines 134-135 accordingly.

Comment 3: Minor typos / change of tense in lines 166, 170, 176, 200 and 227

We thank the reviewer for identifying these. They have all been corrected in the revised manuscript, with the corrections occurring in lines 270, 282, 285, 347 and 383 respectively.

Comment 4: Table 1 should use the same exponent in each cell to enable easier comparisons

Agreed. This change has been made.

Please also note the supplement to this comment:
https://www.ann-geophys-discuss.net/angeo-2019-112/angeo-2019-112-AC2-supplement.pdf

[Figure]

**Supplement:**

[revised manuscript text omitted]